# Evolutionary Trend Analysis of Research on 5-ALA Delivery and Theranostic Applications Based on a Scientometrics Study

**DOI:** 10.3390/pharmaceutics14071477

**Published:** 2022-07-15

**Authors:** You Zhou, Mulan Mo, Dexu Luo, Yi Yang, Jialin Hu, Chenqing Ye, Longxiang Lin, Chuanshan Xu, Wenjie Chen

**Affiliations:** 1Key Laboratory of Molecular Target & Clinical Pharmacology and the State & NMPA Key Laboratory of Respiratory Disease, School of Pharmaceutical Sciences & The Fifth Affiliated Hospital, Guangzhou Medical University, Guangzhou 511436, China; 2020310799@stu.gzhmu.edu.cn (Y.Z.); 2019218295@stu.gzhmu.edu.cn (M.M.); 2020519509@stu.gzhmu.edu.cn (D.L.); 2021210828@stu.gzhmu.edu.cn (Y.Y.); 2021210827@stu.gzhmu.edu.cn (J.H.); 2Fujian Province University Key Laboratory of Green Energy and Environment Catalysis, College of Chemistry and Materials, Ningde Normal University, Ningde 352100, China; cqy@ndnu.edu.cn; 3Shenzhen Osteomore Biotechnology Co., Ltd., Shenzhen 518118, China; linlx@hotmail.com; 4State Key Laboratory of Respiratory Disease, Guangdong-Hongkong-Macao Joint Laboratory of Respiratory Infectious Disease, Guangzhou 510182, China; 5Sydney Vital Translational Cancer Research Centre, Westbourne St., Sydney, NSW 2065, Australia

**Keywords:** 5-aminolevulinic acid (5-ALA), nanotechnology, bibliometric analysis, CiteSpace

## Abstract

5-aminolevulinic acid (5-ALA) has been extensively studied for its sustainability and broad-spectrum applications in medical research and theranostics, as well as other areas. It’s a precursor of protoporphyrin IX (PpIX), a sustainable endogenous and naturally-existing photosensitizer. However, to the best of our knowledge, a scientometrics study based on the scientific knowledge assay of the overall situation on 5-ALA research has not been reported so far, which would be of major importance to the relevant researchers. In this study, we collected all the research articles published in the last two decades from the Web of Science Core Collection database and employed bibliometric methods to comprehensively analyze the dataset from different perspectives using CiteSpace. A total of 1595 articles were identified. The analysis results showed that China published the largest number of articles, and SBI Pharmaceuticals Co., Ltd. was the most productive institution that sponsored several of the most productive authors. The cluster analysis and burst detections indicated that the improvement of photodynamic efficacy theranostics is the up-to-date key direction in 5-ALA research. Furthermore, we emphatically studied nanotechnology involvement in 5-ALA delivery and theranostics research. We envision that our results will be beneficial for researchers to have a panorama of and deep insights into this area, thus inspiring further exploitations, especially of the nanomaterial-based systems for 5-ALA delivery and theranostic applications.

## 1. Introduction

5-aminolevulinic acid (5-ALA) is a natural non-protein amino acid that exists as a common precursor of heme in animals [1]. Endogenous 5-ALA is synthesized from succinate coenzyme A and glycine catalyzed by the 5-aminolevulinate synthase. Afterward, 5-ALA follows the metabolic pathway for protoporphyrin IX (PpIX) synthesis, and finally PpIX chelates with ferrous ions to produce heme, which is the rate-limiting step [2]. Interestingly, 5-ALA formation in cells is tightly controlled by heme through a negative feedback-mediated regulation of 5-aminolevulinate synthase [3,4]. Due to the miraculous properties of PpIX and heme, exogenous 5-ALA was widely investigated as a prodrug in medical fields and has already been clinically utilized [5]. 5-ALA was firstly approved by the European Medicine’s Agency in 2007 as an optical imaging agent in surgery to visualize the malignant tissue, then approved by the U.S. Food and Drug Administration for the same application in 2017 [6].

5-ALA has been a star molecule with excellent medical applications. Mechanistically, exogenous 5-ALA utilization can induce a rapid formation and accumulation of PpIX due to the deficiency of iron chelatase activity in tumor tissues [7,8]. Owing to its unique photosensitive activities, when irradiated by a specific light source, PpIX can emit fluorescence that can be used to discriminate the malignant tissues from normal tissues [9,10,11]. For example, the assistance of 5-ALA-derived PpIX fluorescence enables more complete resections of tumors, largely improving the progression-free survival in glioma patients [12,13]. Additionally, PpIX irradiated by light or ultrasound can produce reactive oxygen species (ROS), which induce lethal oxidative damage to membrane lipids, nucleic acids, and proteins. Therefore, 5-ALA assisted photodynamic/sonodynamic therapy provided non-invasive alternatives for tumor treatment. Critically, the multitargets of ROS oxidation to biological macromolecules make it hard for tumors to develop resistance to photodynamic or sonodynamic therapies [14,15]. In light of safety and efficacy, 5-ALA has been approved as a photodynamic therapy agent against several tumors and bacterial infections [5,15,16,17,18,19,20]. Apart from the PpIX-associated benefits, 5-ALA may be used to prevent mitochondrial-related metabolic diseases because an insufficiency of 5-ALA will directly result in mitochondrial dysfunction, which can be the direct cause of mitochondrial-related metabolic diseases [21,22,23]. Beyond that, it can be sustainably applied in many other areas as well. In studies of livestock feeding, a 5-ALA addition in forage was found to be beneficial to the animal’s immune performance and their growth [24,25]. Other studies also showed that 5-ALA in low concentrations can enhance plant tolerances to salinity, drought, heat, coldness, and other abiotic stresses [26,27]. Nowadays, more and more eco-friendly methods, such as metabolic engineering to produce 5-ALA, have been rationally designed and developed to overcome the shortcomings of chemical synthesis [28]. Methyl ester aminolevulinic acid and hexyl ester aminolevulinic acid, two derivatives of 5-ALA, have also gained their marketing authorization to overcome the limited local bioavailability of 5-ALA in daily clinical practice [29]. With the increasing development of 5-ALA synthesis, it can be foreseen that 5-ALA could have broader prospects. Despite the considerable success that has been achieved during the history of exploring 5-ALA, there are still challenges to be addressed and opportunities to be explored. The main issue among these obstacles is how to manipulate the targeting ability of 5-ALA to control its biological distribution and intracellular accumulation, which could improve the efficacy of 5-ALA and extend its ranges of applications [30].

Scholars have devoted themselves to the research of 5-ALA, boosting vast numbers of publications. These publications offer tremendous information, which might be too miscellaneous for the readers to grasp the major thread. To this aim, it is urgent to depict the panorama of the 5-ALA research field. Although some reviews have summarized the basic information and latest progress of this field for readers, neither new perspectives have arisen from the whole research ecosystem nor have quantitative analyses been previously reported [31,32]. For this purpose, since bibliometric analysis could unearth the complex information in a visualizable way, the operation of metrology software was carried out to obtain the present valuable contents and future trends of 5-ALA research. In this study, we tried to map the basic knowledge structure of this field, figuring out the hotspots and frontiers with the assistance of the dataset and software. After a comprehensive analysis, we investigated the participation and contributions of different countries, institutions, journals, and authors. Furthermore, we determined the present main topics and future directions, which could help researchers who are devoted or interested in the applications of 5-ALA.

## 2. Material and Methods

### 2.1. Retrieval Strategy and Data Collection

Web of Science (WoS, Clarivate Analytics, Philadelphia, PA, USA), one of the largest academic information databases, was used as the source to collect the related publications for the following study. Specifically, the Science Citation Index Expanded (SCI-Expanded, 2006–present) of WoS Core Collection (WoSCC) was selected in order to gather a full collection. The retrieval formula was constructed as follow: TI = (*aminolevulinic acid OR 5 ALA), and only articles were included. Afterwards, the results were refined for publication years between 2006 and 2021 to get rid of the bias due to daily updates. Ultimately, 1595 publications were found. The distribution information of these publications for different countries, institutions, grants, journals, authors, and their annual citations were acquired in WoS by its analysis reports for graphing, while the retrieval records were also exported to plain text files in the format of full record and cited references for subsequent in-depth analysis.

### 2.2. Data Selection and Analysis

Microsoft EXCEL 2016 (Microsoft, Redmond, WA, USA), GraphPad Prism 5.0 (GraphPad Software, San Diego, CA, USA), and CiteSpace 5.8.R3 (Chaomei Chen, Beijing, China) were employed to run the statistical tests and graph plotting; the complex hidden information was mainly captured using CiteSpace. Briefly, the dataset was transformed to CiteSpace-recognizable files, as per the user guide, before running the analysis projects. Specifically, time slicing was set as years per slice and the selection criteria was termed as the g-index (cumulative citation number multiplied by 25). The term source can be extracted from all options, while node types should be checked each time to be in line with corresponding objects. Links and pruning were set up in cosine with slices and pathfinder, respectively. After the visualization run, the results were available as a cluster map containing colored nodes and strings. To refine the results, nodes with the same meaning and string difference were merged, whereas meaningless nodes were deleted. The built-in LLR method was used as a standard when performing the cluster analysis. As well, all the modularity values (Q-value) were > 0.3 and the mean silhouette values (S-value) were > 0.7, which can be considered a significant clustering in the showed results [33]. The bibliometric analysis process is summarized in the flow chart below (Figure 1A).

## 3. Results and Discussion

### 3.1. General Information of Publications

Basically, the publications are wildly accepted as the referable index for showcasing a specific research field, especially when delivering a message in the form of visual post-multivariate analyses [34]. To understand the developments and research trends of 5-ALA over the past two decades from a macro-perspective, the timespan ranging from 2006 to 2021 was retrieved in WoSCC for the experiments. After screening the literature through fixed processing, it hit 2159 records including 1595 articles and 614 other document types. In the histogram for annual publication, the number of publications increased steadily from 2006, with small fluctuations between 2011 and 2012. However, when it comes to the year of 2019, the growth rate of publications increased significantly compared with past years (Figure 1B and Appendix A). As well, the cumulative citations were 29,790, with an average of 18.88 citations per publication. Meanwhile, the citations also showed a steady acceleration in 2019 (Figure 1B). The trends of both publications and their citations suggest that the research of 5-ALA has increasingly developed with a steady impetus in recent years.

### 3.2. Knowledge Structure Features of 5-ALA Research

#### 3.2.1. Analysis of the Contribution of Countries

There are 56 countries participating in the exploration of 5-ALA properties, and Figure 2A shows the geographical distribution. As shown in Table 1, the major research countries ranked by the publication of items are China (445 papers), Japan (325 papers), the US (246 papers), and Germany (147 papers). The number of articles they published were far more than other countries, with these four countries contributing 72.87% of all the publications from a total number of 1163 articles. Consistently, China, Japan, the US, and Germany are the most cited countries with 6763, 4986, 4858, 6314 citations, respectively, which account for 76.94% of all the citations. Apart from the gross volume, China also maintained the most growth in publications among these highly productive countries (Figure 2B). However, in terms of average citations per item (ACI), a symbolic indicator reflects the value and contribution of one paper to the scientific community [33], Germany tops this category with 42.90 citations per paper (Table 1), which is more than two-fold higher than the average value of 18.88. It is indicated that these four countries were more active in exploring the 5-ALA-related field of research than any other country, and they achieved more valuable discoveries that deserved publishing. 

To further mine the key contributions of each country, we performed a CiteSpace analysis to identify the roles they played in this area. In a co-occurrence analysis of countries and regions (Figure 2C), not only were the countries with a high frequency of publication visually shown by the area of the representative circles, but they were also exhibited by the number changes with year-lapse by constructing the circles in a multi-layered concentric ring. What’s more, countries have been linked with colored lines, which represent cooperation between different countries. Specifically, connections with a warm (red) color represent close collaborations, while the colder color (pale) shows moderate collaboration. Additionally, for the co-occurrence detection, the betweenness centrality, which measures the number of times a node lies on the shortest path between other nodes in the network, would be used to weigh the importance of nodes within whole networks [35]. The outermost purple ring around a circle disclosed the central intermediation information. The thicker the purple rings are, the higher the betweenness centrality will be. The results revealed that England (centrality of 0.6), France (centrality of 0.49), Australia (centrality of 0.43), Switzerland (centrality of 0.41), and the Netherlands (centrality of 0.31) are the countries with a higher centrality compared to other countries, which means they have played major roles in research and maintained a high degree of cooperation with other countries and regions—although they would not be that outstanding in publication numbers. On the contrary, those countries with large numbers of publications but small centralities, such as China (centrality of 0.09), the US (centrality of 0.17), and South Korea (centrality of 0.00), seemed less collaborative to others.

#### 3.2.2. Analysis of the Contribution of Organizations

With respect to research institutes, WoSCC identified that 1639 institutions in total contributed to the research of 5-ALA. It is usually hard to describe all their contributions in detail, therefore, here, we have figured out the top-10 most prolific institutions and demonstrated their achievements in the aspects of publication counts, ACI, and *h*-index (Figure 2D). Even though it started (2013) seven years later than second place (Fudan University, 2006), SBI Pharmaceuticals Co., Ltd. has the most publications, which nearly doubled second place—while they have a similar *h*-index. As well, the Heinrich Heine University, Düsseldorf, has got the highest ACI value of 103.64; specifically, 28 publications have been cited 2902 times. It suggests that they have done some fundamental work that has received extensive attention in this area. After careful retrieval, we found that the highest ACI was almost attributed by a randomized, controlled multi-centre phase-III trial. In a surgery that involved the resection of malignant glioma in the clinical trial, 5-ALA-fluorescence guided imaging enabled more complete resections by enhancing tumor contrast, therefore improving progression-free survival successfully. Until 2021, it has been cited 2032 times.

Similar with the co-occurrence analysis of countries and regions, the co-occurrence frequency matrix of institutions can be used as an index to describe cooperation, which is denoted when two authors’ institutions appear in one article. Figure 2E exhibits co-institutes in the field of 5-ALA. The publication numbers and the co-occurrence frequencies of institutes can be referred to by the size of their circles and labels respectively. Additionally, the linkage lines between two nodes illustrate the existence of cooperation. A thicker line implies a stronger partnership, while the color indicates the time when they began to cooperate. Of note, there is no obvious centralized node, and partial institutions conducted almost no collaborations with others. However, institutions that made major contributions have formed a network, such as the linkage within SBI Pharmaceuticals Co., Ltd., the Tokyo Institution of Technology, Fudan University, and Harbin Medical University, who all established an intimate collaboration.

The support from funding agencies is indispensable to the research accomplishments. To this end, the most active funding agencies in this field were also assessed (Appendix A). The data showed the top 5 of 668 institutions who shared 24.69% of all the publications, which suggests that publications in the field of 5-ALA were specifically contributed to from only several agencies. Moreover, when comes it to the nationality of these organizations, the conclusions will be in agreement with the abovementioned that China, Japan, the US, and Germany have made greater efforts in exploring the issues of 5-ALA than other countries.

#### 3.2.3. Analysis of the Contribution of Journals

The international peer-reviewed journals are the well-established platforms for scientific publications and science communication. An analysis of journal distribution based on publications will depict the current map of journals and offer critical perspectives. Statistically speaking, these 5-ALA-related papers were published by 613 journals. Table 2 summarizes the top-11 most active journals. Papers published in the journal *Photodiagnosis and Photodynamic Therapy,* accounted for the highest proportion of all related papers; to be specific, 10.34% (165 in 1596) of the publications were reported by this journal, followed by the *Journal of Photochemistry and Photobiology B: Biology* (38), the *Journal of Neurosurgery* (31), *Lasers in Surgery and Medicine* (28), and *PLoS ONE* (25). Four-elevenths of them were in the top 25% ranking of similar journal categories (Q1), and four-elevenths were in the 25–50% ranking (Q2). Importantly, the ACI of 5-ALA-related literature were higher than the journal’s expected average citations, especially the papers published in the *Journal of Neurosurgery*, whose ACI reached 41.97. These data suggest that 5-ALA-related papers have received more attention than other topic papers in the same journals. Additionally, the top-11 most active journals are sponsored in developed countries, with 6 of them being issued in the US and the others originating from Europe [36]. However, the productive countries like China, Japan, and South Korea were absent from the publishing journals. The founding and development process of an international journal needs a large number of resources, which apparently will be affected by the economy. So, the reason for productive journals mainly coming from western countries may be because developing countries do not have enough resources to found and operate international journals.

Journals can be categorized into different subjects for clarifying research scopes. These categories may help the overall understanding of 5-ALA. Figure 3A presents the top-10 subject categories according to the publication numbers in the WoSCC database. Oncology, Surgery, Biochemistry Molecular Biology, Dermatology, and Clinical Neurology were the most intensive subjects in the studies of 5-ALA. Furthermore, a co-citation analysis was performed to demonstrate the influences and relationships between journals. As shown in Figure 3B, there were numerous journals with high citations but low betweenness centrality, which is referenced by the circle area and purple ring, respectively. The colors of connecting lines denote the time of co-citation occurrence, which can be referred to in the legend of rainbow bars colored from purple to yellow. Therefore, the purple lines gathered in the 21 research topics’ complex network indicated that these co-citations occurred in the early years during the time span of 2006–2021. While the journals including *Nature*, *Applied Microbiology and Biotechnology*, and *Scientific Reports* in the cluster of “metabolic engineering” is newly emerged.

#### 3.2.4. Analysis of the Contribution of Influential Authors

Worldwide scholars devoted to this field have done concrete studies. Given this, it is important to analyze this subject in terms of authors. There are more than 6000 authors that have taken part in studying 5-ALA, and Figure 3C shows the top 11 productive authors with a detailed *h*-index and ACI. Tanaka, T. was the author with the highest publication numbers, followed by Nakajima, M., Takahashi, K., Inoue, K., and Ishizuka, M. Interestingly, all five authors were from Japan. As well, they were all affiliated with SBI Pharmaceuticals Co., Ltd., except Inoue, K. Whereas, Inoue, K. has the largest *h*-index of all of them. Another noteworthy point is that Stummer, W. has a super high ACI value of 109.97. These high citations can mainly be attributed to the work of his team on the clinical diagnostic and therapeutic value of 5-ALA, given that his team focused on the applications of 5-ALA on glioma for more than 10 years [12,13,37,38]. Meanwhile, Stummer, W. holds the first, second, and fifth places in the list of top-10 most-cited literature (Table 3).

The co-authorship information is beneficial for the readers to acquire details of the authors’ activities and to learn about their whole team’s achievements. The co-authorship analysis in this study visually showed the publication numbers of each author by the circle size. As well, the map clearly shows the major cooperative teams more than the research teams from SBI Pharmaceuticals Co., Ltd. by their co-authorship network (Figure 3D). Besides, co-citation is denoted as a relationship where two authors appear together in the reference list of a third document, which is often used to reveal the key authors in subdivided areas. As displayed in Figure 3E, the co-citation network was divided into several subareas. However, the betweenness centrality values of the authors are quite small, demonstrating few influential authors were occupying multiple subareas (Appendix A). From the above analytic results, it can be assumed that Tanaka, T., Inoue, K., and Stummer, W. are the most influential authors.

### 3.3. Overview of Hotspots and Beyond

#### 3.3.1. Characteristic of Research Topics for 5-ALA

Keywords are extracted from papers to express the core ideas, which may include the key information of research topics, methods, etc. For bibliometrics, a keywords analysis is the most powerful way to accurately excavate some underlying messages. Firstly, the frequencies of keywords can directly reflect their prevalence in present studies. Secondly, the co-occurrence of two keywords that appear in one paper will provide a close association between different contents. Thirdly, the keywords-formed clusters could point out the composition of main topics and their evolutionary trend. Table 4 shows the top-30 keywords sorted by their frequencies. These keywords mainly belong to molecular biology, oncology, and surgery. However, as inferred from keywords like “photodynamic therapy”, “protoporphyrin ix”, “fluorescence”, and “cancer” ranking in the front, we can conclude that the applications of 5-ALA in cancer treatment should be the most popular research direction. As well, it means that they have propped up the main development for 5-ALA. More than 400 keywords were collected for the analysis of co-occurrence in this study. These larger circles have formed a complex network, demonstrating that 5-ALA in cancer treatment has been studied in more depth than other research directions (Figure 4A). Nevertheless, keywords such as “inhibition”, “oxidative stress”, “malignant glioma”, and “porphyrin” have a large centrality within 767 pairs of connection (Figure 4A). Additionally, the keywords can be clustered into 19 subareas with distinct topics. Figure 4B showed the evolutionary trend of the main topics and their interactions in a timeline format. On the basis of the co-cited times represented by circle areas and their distribution density, “ppix”, “photodynamic therapy”, “photodynamic diagnosis”, and “actinic keratosis” were the most attractive topics in the field. In other words, these topics gathered a high intensity of research efforts. It is worth noting that the majority of keywords with high citations appeared in the early period of the timespan but were seldom proposed after 2010. Presumably, there were fewer original innovations and brand-new breakthroughs emerging during the recent decade. Since most of the popular keywords and clusters were related with photodiagnosis and photodynamic therapy, we have drawn the annual changes for photodiagnosis/photodynamic-therapy-related publications (Figure 4C). Interestingly, 911 publications were defined, and its growing path seems similar to the growth of total publications from 2019 compared with the past years, indicating that the major growth of publications in this area in the last three years were from the photodiagnosis/photodynamic-therapy-related studies (Appendix A). Given the conception that photodiagnosis and photodynamic therapy has been a common description for 5-ALA applications for years [17], we have performed a second-level analysis after shielding the common keywords associated with photodiagnosis and photodynamic therapy (Appendix A). The produced landscape showed that the clusters mainly associated with the applications of 5-ALA including “derivative”, “in vitro”, “oxidative stress”, “photosynthesis”, “diabetes”, and “sonodynamic therapy” were newly emerging. Nonetheless, some major issues hindered the development of current and further potent applications, such as the instability of the active pharmaceutical ingredients, poor tissue penetration, finite normal-to-lesion tissue selectivity, and photosensitization reactions from systemic drug absorption. To address these issues, especially the efficient delivery of 5-ALA to specific sites for on-demand theranostics, scientists are seeking other reliable techniques such as chemical modifications and smart drug delivery carriers [41,42,43].

#### 3.3.2. Research Hotspots and Frontiers of 5-ALA

Given that the abovementioned assays of the co-occurrence network of keywords shows few keywords emerging during last decade, it’s critical to compute the most cutting-edge directions that are being focused on in this area. Actively discussed conceptions can reflect the rise or transfer of research interests, which may also portend the potential beginning of a new scientific subarea. Burst detections can perfectly fulfill the demand of determining the sharply increased popularity. Since the shortage of new influential conceptions in keywords co-citation analysis, we chose to detect the burst items in this field. Figure 5A illustrates the top-25 strongest keywords of citation burst. The photodynamic diagnosis and therapy-related keywords such as “topic application”, “endogenous protoporphyrin”, “lead”, and “photosensitization” simultaneously burst in early time. However, in recent years, keywords like “resistance”, “impact”, “pathway”, and “system” emerged steadily. It can be inferred that the research hotspots of 5-ALA shifted from the clinical applications to the efficacy improvement and fundamental biological effects. 

In terms of authors, the results suggest that the latest burst authors are “Nakajima, M.”, “Tanaka, T.”, “Widhalm, G.”, “Li, X.K.”, and “Zeng, K.” (Figure 5B). As mentioned above, Nakajima, M. and Tanaka, T. were apart of the same team, with highly-cited works that focused on the cancer theranostics of 5-ALA. Particularly, their team revealed the pivotal roles of peptide transporter PEPT1 and ATP-binding cassette transporter ABCG2 in regulating intracellular PpIX levels and determining the efficacy of 5-ALA-based photocytotoxicity against gastric cancer cells, because these two transporters were found to mediate the influx of 5-ALA and efflux of PpIX, respectively [44]. Another researcher, Li, X.K., who is a close collaborator with Nakajima, M. and Tanaka, T., mainly concentrated on the biological effects of 5-ALA-induced heme oxygenase-1 in diseases [45,46]. Dr. Widhalm is a productive author who works on glioma therapy. Importantly, he proposed that 5-ALA-induced fluorescence can be a powerful marker for glioma histopathological grading, and can even be a prognosis marker [40,47,48]. As for professor Zeng, he is a major contributor to 5-ALA-based photodynamic therapy against human papillomavirus infection. His research may have accelerated the development of photodynamic therapy against virus infections [49,50,51]. By combing cluster analysis and burst detection, the results indicate that 5-ALA-mediated photodiagnosis and photodynamic therapy will dominate the 5-ALA-related field from now on.

#### 3.3.3. Nanotechnology Is a Potential Rising Star

The photodiagnosis and photodynamic therapeutic abilities of 5-ALA were attributed to its PpIX-mediated photosensitive activities. By acting as a photosensitizer, the efficacy of 5-ALA was restricted by its uptake and accumulation at specific locations [5]. Numerous studies have proven that photosensitizers could be remarkarbly upgraded with better efficiencies under nanotechnological assistance, either in terms of physicochemical properties or biological activity [52]. For instance, laboratory research indicated that the employment of poly-(lactic-co-glycolic acid) or metal-based nanoparticles as delivery carriers enhanced the uptake of 5-ALA and the following PpIX conversion, which therefore promoted its efficacy [53,54,55,56]. In a registered 5-ALA gel, 5-ALA was loaded into a nanoscale-lipid vesicle for the treatment of actinic keratosis. This nano-emulsion formulation stabilized 5-ALA and enhanced its penetration through the stratum corneum [57]. In the meantime, nanocarriers modified with different properties could enhance the selectivity and accumulation of 5-ALA. When embedded 5-ALA is in hypoxia-responsive amphiphilic polymer nanovesicles, it would specifically release at the tumor site when stimulated by a hypoxic tumor microenvironment [58]. While functionalized 5-ALA loaded hollow mesoporous silica nanoparticles with folic acid ligands, the nanoparticles can assist 5-ALA in bypassing the lipophilic barrier to directly target cancer cells [59]. There are also attempts to increase the efficacy of 5-ALA by introducing other promising drugs with advantageous delivery systems. These strategies could overcome the issue of oxygen deficiency in photodynamic therapies and/or synergistically augment their cytotoxicity to cancer cells [58,60].

The representative practices above inspired us to incorporate nanodelivery systems, which will have rising propensity in the field of 5-ALA research and applications. To further interpret the contemporary profile of nanotechnological involvement, we profiled the status of nano-system applications in 5-ALA. While PpIX, as the active form of 5-ALA, were also included in the profiles. Figure 6A shows that the publications were in an upward trend. As well, it’s comparable with the publication numbers of photodiagnosis/photodynamic-therapy-related research (Appendix A). Additionally, the main research topics in these publications were visualized by a keywords cluster analysis. The results indicate that these studies can be approximately divided into three directions (Figure 6B). The first illustrates its basic applications and mechanisms, such as “ppix”, “cytotoxicity”, “in vitro”, “bladder cancer”, and “photodynamic/photothermal therapy”; the second is about nanocarriers, including “nanoparticles” and “carbon nanotube”; and the third main direction focuses on the field of pharmacokinetic characteristics, which is composed of “release” and “sustained release”. These clusters may supply inspiration to fellow researchers because they demonstrate popular directions.

## 4. Limitations

Some limitations of this study should be acknowledged. This work was based on a dataset from public databases, with biases resulting from confounding factors that might exist. In addition, we made conclusions based on the published articles. As such our strategy has ignored those works published as patents, which may be also very important for research and development. However, WoSCC was the most-used database for scientmetrics research, and it gathered different journals from all over the world, which ensures our present study still rationally reflects the whole scenario of this area.

## 5. Conclusions

In this study, we have conducted a systematic bibliometric analysis of 5-ALA research between 2006 and 2021. The data indicate that 5-ALA has attracted more and more attention from scientists. In the process of attainment in this research field, China, Japan, the US, and Germany are the major countries that contributed in different dimensions. China has the largest number of publications and citations, while Japan fostered highly-productive authors. The US has excellent performance in supporting funds and publishment, and Germany firstly launched the clinical applications of 5-ALA. So far, SBI Pharmaceuticals Co., Ltd. and *Photodiagnosis and Photodynamic Therapy* were the most active institution and journal, respectively. Of note, Stummer, W. is the most influential author. According to the cluster analysis and burst detection, the hotspots in this field are mainly concentrated on photodiagnosis and photodynamic therapy, while its derivatives and their fundamental biological effects were also emerging. Additionally, we have emphatically investigated the present applications of nano-systems that have been implemented in 5-ALA-related research. Overall, this study provides the knowledge structure map, evolution trend, and the frontiers of 5-ALA research. It is anticipated that this work could offer a forum for the scientific community to maximize productivity through the creative assembly of expertise from different directions.

## Figures and Tables

**Figure 1 pharmaceutics-14-01477-f001:**
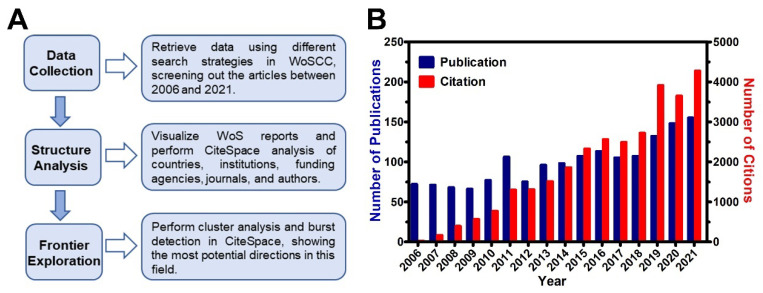
The flow chart of bibliometric analysis and basic profile of 5-ALA research. (**A**) The flowchart of bibliometric analysis mentioned in methods section. (**B**) The annual distributions of 5-ALA publications and their citations.

**Figure 2 pharmaceutics-14-01477-f002:**
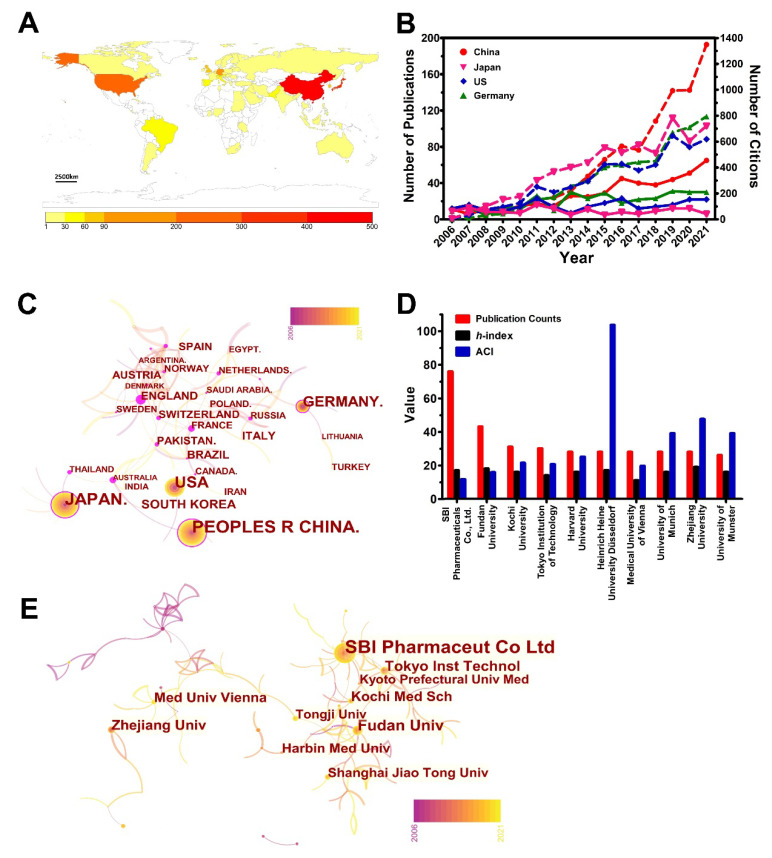
Contributions of different countries and institutions. (**A**) The exhibition of countries in the geographical map with publication counts. (**B**) The evolution trends of publications and citations of four major countries. Solid line indicates publications and dotted line indicates citations. (**C**) Co-occurrence network of countries analyzed by CiteSpace. (**D**) The graph of publication number, *h*-index, and ACI of the top-10 most active institutions. (**E**) Cooperation network of institutions analyzed by CiteSpace.

**Figure 3 pharmaceutics-14-01477-f003:**
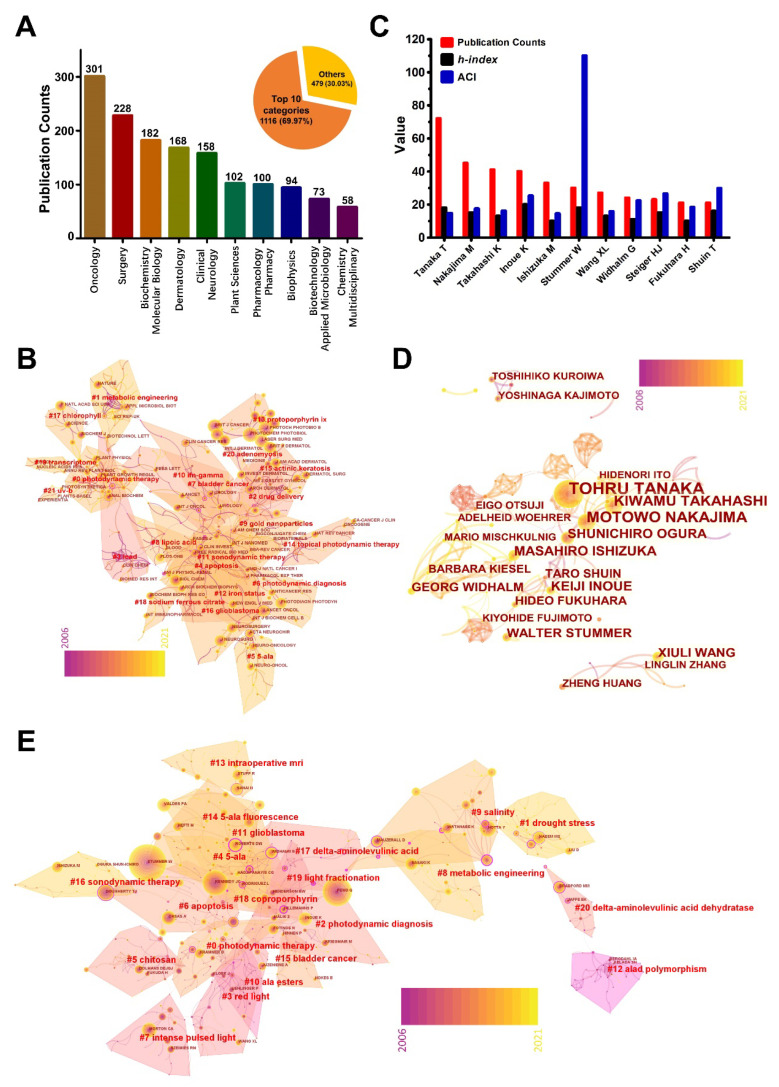
The analysis of 5-ALA research publishers and authors. (**A**) The numbers for top-10 most relevant subject categories for publications. (**B**) Co-citation network of journals analyzed by CiteSpace. (**C**) The publications, *h*-index, and ACI of top-11 productive authors. (**D**) Cooperation network of authors analyzed by CiteSpace. (**E**) Co-citation network of authors analyzed by CiteSpace.

**Figure 4 pharmaceutics-14-01477-f004:**
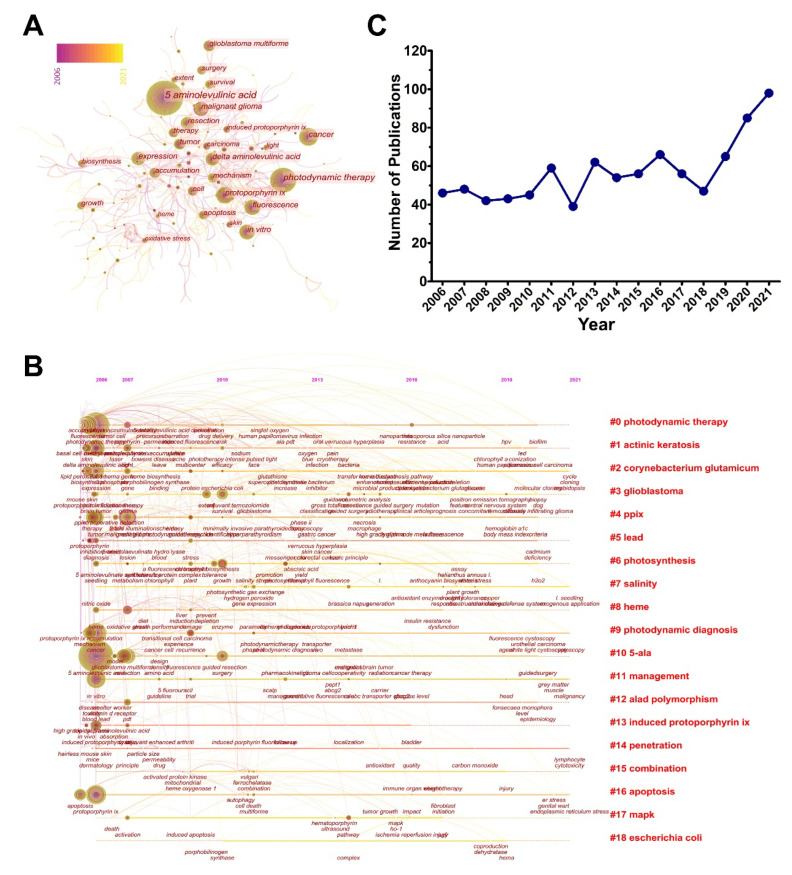
Research topics of 5-ALA research related photodiagnosis/photodynamic therapy. (**A**) Co-occurrence network of keywords in this field analyzed by CiteSpace. (**B**) Major research topics clustered by CiteSpace. (**C**) The annual changes of 5-ALA based photodiagnosis/photodynamic therapy publication. Data were achieved in the retrieve formula of TI = (*aminolevulinic acid OR 5-ALA) AND TS = (photodiagnosis OR photodynamic) and articles only.

**Figure 5 pharmaceutics-14-01477-f005:**
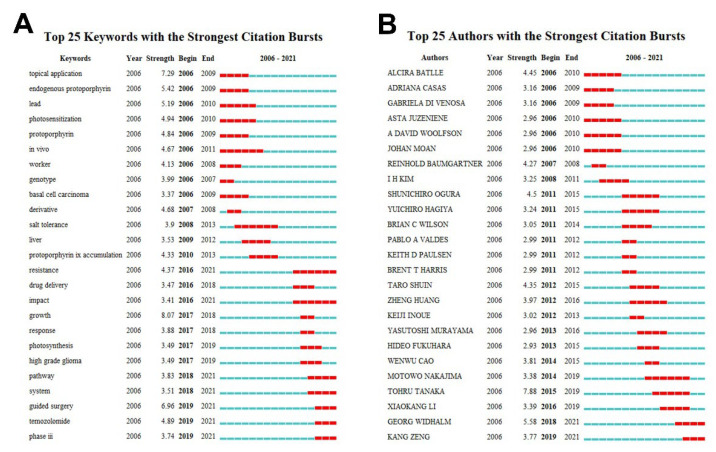
Hotspots and frontiers in 5-ALA research field. (**A**) The top-25 keywords with strongest citation burst detected by CiteSpace. (**B**) The top-25 authors with strongest citation burst detected by CiteSpace.

**Figure 6 pharmaceutics-14-01477-f006:**
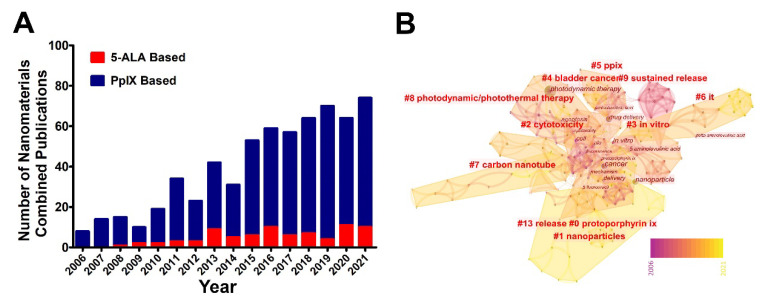
Key information for nanomaterials combined 5-ALA research. (**A**) The trends of 5-ALA-based and PpIX-based annual publication numbers. Data were collected in the retrieval formula of TI = (*aminolevulinic acid OR 5 ALA) and TS = (nano*), and TS = (Protoporphyrin ix OR PpIX) and TS = (nano*) respectively, publications were articles only. (**B**) Cluster analysis diagram of nanomaterials combined 5-ALA research by CiteSpace.

**Table 1 pharmaceutics-14-01477-t001:** The information of the top-20 countries according to the publication analysis in 5-ALA research.

Rank	Country	Between Centrality	Publications	% of 1595	*h*-Index	Times Cited	ACI
1	China	0.14	445	27.88	39	6763	15.05
2	Japan	0.19	325	20.36	35	4986	15.17
3	United States	0.05	246	15.41	36	4858	19.52
4	Germany	0.14	147	9.21	37	6314	42.50
5	South Korea	0.00	69	4.32	18	907	12.00
6	United Kingdom	0.65	64	4.14	22	1252	19.56
7	Brazil	0.19	51	3.19	18	833	16.33
8	Italy	0.05	44	2.76	14	611	13.89
9	Switzerland	0.37	40	2.51	18	1158	28.95
10	Pakistan	0.31	38	2.38	20	1299	34.18
11	Austria	0.01	36	2.26	14	859	23.86
12	Spain	0.08	35	2.19	14	641	18.31
13	Norway	0.14	28	1.75	12	526	18.79
14	Netherlands	0.23	26	1.63	16	630	24.23
15	Canada	0.14	24	1.51	14	819	34.13
16	France	0.54	24	1.51	10	253	10.54
17	Poland	0.05	24	1.51	8	168	7.00
18	Turkey	0.00	22	1.38	8	214	9.73
19	Iran	0.00	21	1.32	8	204	9.71
20	Russia	0.22	21	1.32	8	175	8.33

**Table 2 pharmaceutics-14-01477-t002:** Top 11 most active journals in the field of 5-ALA research.

Journal	Publisher’s Country	Publication Counts	ACI	Categories	Quartile in Category	5-Year Impact Factor	*h*-Index
*Photodiagnosis and Photodynamic Therapy*	Netherlands	165	8.73	Oncology	3	3.60	19
*Journal of Photochemistry and Photobiology B: Biology*	Switzerland	38	16.08	Biophysics; Biochemistry & Molecular Biology	1	5.38	16
*Journal of Neurosurgery*	United States	31	41.97	Clinical Neurology; Surgery	1	5.15	18
*Lasers in Surgery and Medicine*	United States	28	22.25	Dermatology; Surgery	1	3.38	14
*PLos One*	United States	25	28.76	Multidisciplinary Sciences	2	3.79	15
*Photochemistry and Photobiology*	United States	23	23.00	Biophysics; Biochemistry & Molecular Biology	2; 3	3.06	15
*Photodermatology Photoimmunology & Photomedicine*	Denmark	23	13.96	Dermatology	2	3.16	12
*Scientific Reports*	England	22	15.00	Multidisciplinary Sciences	1	5.13	10
*World Neurosurgery*	United States	21	10.38	Surgery; Clinical Neurology	3; 4	2.32	7
*Acta Neurochirurgica*	Austria	20	28.30	Surgery; Clinical Neurology;	3; 4	2.47	11
*Journal of Neuro-Oncology*	United States	20	19.25	Clinical Neurology; Oncology	2; 3	4.23	11

**Table 3 pharmaceutics-14-01477-t003:** Top-10 most cited literatures in the field of 5-ALA.

Title	First Author	Journal	Publication Year	Total Citations
Fluorescence-guided surgery with 5-aminolevulinic acid for resection of malignant glioma: a randomised controlled multicentre phase III trial [12]	Stummer, Walter	*Lancet Oncology*	2006	2032
Counterbalancing risks and gains from extended resections in malignant glioma surgery: a supplemental analysis from the randomized 5-aminolevulinic acid glioma resection study Clinical article [13]	Stummer, Walter	*Journal of Neurosurgery*	2011	198
Clinically relevant reduction in risk of recurrence of superficial bladder cancer using 5-aminolevulinic acid-induced fluorescence diagnosis: 8-year results of prospective randomized study [10]	Denzinger, Stefan	*Urology*	2007	198
Co-registered fluorescence-enhanced tumor resection of malignant glioma: relationships between delta-aminolevulinic acid-induced protoporphyrin IX fluorescence, magnetic resonance imaging enhancement, and neuropathological parameters Clinical article [11]	Roberts, David W	*Journal of Neurosurgery*	2011	190
5-Aminolevulinic Acid-derived Tumor Fluorescence: The Diagnostic Accuracy of Visible Fluorescence Qualities as Corroborated by Spectrometry and Histology and Postoperative Imaging [37]	Stummer, Walter	*Neurosurgery*	2014	171
Photodynamic therapy of acne vulgaris using 5-aminolevulinic acid versus methyl aminolevulinate [18]	Wiegell, Stine Regin	*Journal of the American Academy of Dermatology*	2006	151
Study on physicochemical properties of ionic liquids based on alanine [C(n)mim] [Ala] (*n* = 2, 3, 4, 5, 6) [39]	Fang, Da-Wei	*Journal of Physical Chemistry B*	2008	149
Novel development of 5-aminolevurinic acid (ALA) in cancer diagnoses and therapy [17]	Ishizuka, Masahiro	*International Immunopharmacology*	2011	145
Effects of 5-aminolevulinic acid on oilseed rape seedling growth under herbicide toxicity stress [26]	Zhang, Wufeng	*Journal of Plant Growth Regulation*	2008	141
5-Aminolevulinic Acid Is a Promising Marker for Detection of Anaplastic Foci in Diffusely Infiltrating Gliomas with Nonsignificant Contrast Enhancement [40]	Widhalm, Georg	*Cancer*	2010	134

**Table 4 pharmaceutics-14-01477-t004:** The top-30 keywords with the largest occurrence times.

Rank	Keywords	Counts	Centrality	Rank	Keywords	Counts	Centrality
1	5 Aminolevulinic acid	388	0.00	16	Cell	77	0.09
2	Photodynamic therapy	230	0.00	17	Therapy	74	0.03
3	Cancer	143	0.01	18	Surgery	72	0.03
4	Protoporphyrin ix	124	0.06	19	Survival	68	0.02
5	Expression	123	0.05	20	Growth	67	0.01
6	In vitro	118	0.05	21	Extent	60	0.03
7	Fluorescence	111	0.02	22	Induced protoporphyrin ix	59	0.12
8	Delta aminolevulinic acid	108	0.02	23	Biosynthesis	59	0.11
9	Malignant glioma	107	0.23	24	Carcinoma	56	0.13
10	Resection	105	0.04	25	Skin	50	0.02
11	Tumor	93	0.03	26	Light	47	0.04
12	Accumulation	89	0.11	27	Heme	43	0.03
13	Mechanism	85	0.07	28	Oxidative stress	43	0.25
14	Glioblastoma Multiforme	80	0.00	29	Delivery	38	0.08
15	Apoptosis	78	0.05	30	Diagnosis	38	0.18

## Data Availability

The dataset can be downloaded from WoSCC according to the retrieval criteria mentioned in methods and materials, and raw results are available in Appendix A.

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
