# Peer review of "Evolutionary Trend Analysis of Research on 5-ALA Delivery and Theranostic Applications Based on a Scientometrics Study"

_pharmaceutics, 2022, doi:10.3390/pharmaceutics14071477_

Round 1

Reviewer 1 Report

The manuscript of You Zhou proved quite interesting results of the bibliometric analysis on ALA application. Working in the area, I definitely found new useful information and recommend publication of the manuscript. However, I have several recommendations listed below.

-          The conclusion about different usage of official language of each country as the reason for absence of publishing journals in several countries, in my opinion, does not accurately describe the situation. Nowadays, it is common practice to publish journals in English independently on the official language. The other reasons seem to be the reason for the publishing politics.

-          I recommend including the articles from Table 3 to the References list.

-          It will be informative to add the comparison on the Publication Counts in the Nanotechnology section between ALA and PpIX. The number of PpIX works is almost an order of magnitude larger.

-          In fact, I am disappointed by the uninformative conclusion that the main field of ALA research is photodiagnosis and photodynamic therapy. It is not the hotspots, there is nothing “hot”. It is just the common description for the most often ALA applications over many years; the terms allow distinguishing the ALA topic in the ocean of anticancer strategies and are routinely used by cancer researchers. Much more useful information can be obtained after “masking” the most common keywords and a closer analysis of the second level.

Minors

-          WoS is not the largest but one of the largest databases. Please, correct.

-          Text in Fig. 1A is blind.

-          The Publication number and calculated percent in Table 1 differ from the description in the main text.

-          It is impossible to read any text in many Figures generated by CiteSpace. I offer to think on larger version, for example, in Supplementary Information.

-          The W. Stummer ranks in the list of the most cited articles are different in Table 3 and the main text.

-          Table 4 summarizes not 15 but 30 keywords.

-          Please, thoroughly check using capital letters in the Figures and Tables, including those in Supplementary Information.

Author Response

Dear reviewer,

Thank you for kind consideration of our manuscript and offering the opportunity of major revision. We revised the manuscript according to your suggestions. The comments were addressed point by point, the changes made in the manuscript was highlighted. Please find the detailed responses below, and refer the enclosed file.

1: The conclusion about different usage of official language of each country as the reason for absence of publishing journals in several countries, in my opinion, does not accurately describe the situation. Nowadays, it is common practice to publish journals in English independently on the official language. The other reasons seem to be the reason for the publishing politics.

Response: To settle the above suspicion, we have investigated the top 11 journals, finding that all of them were established more than ten years (more than 37.7 years in average indeed). Having investigated the process of how to sponsor an SCI journal, we learnt that the founding and development of an international journal needs large number of resources and efforts, including the factor of economy status. When taken the economic state of each country in to considerate, the results showed that those 11 journals are all from developed countries [1]. On the other hand, those developing countries may not have enough resources to found and operate international journals decades ago. So, this may be the reason why the productive journals are mainly founded and issued in western countries.

This conclusion was used to replace the official language hypothesis in the revised manuscript at page 11 line 238-244, which has been highlighted in red in the main text, it now reads as below:

“Additionally, the top 11 most active journals are sponsored form the developed countries, six of them were issued in the US and the others were originated from Europe. But the productive countries like China, Japan, and South Korea were in the absence of publishing journals. The founding and development process of an international journal needs large number of resources, which apparently will be affected by the economy. So, the reason for productive journals mainly coming from western countries may be because developing countries do not have enough resources to found and operate international journals decades ago.”

2: recommend including the articles from Table 3 to the References list.

Response: Thanks for your kind reminder, we have now included the relevant articles into the reference in the revised manuscript. These are all highlighted in red in the main text, also can be seen below:

  1. Denzinger S, Burger M, Walter B, Knuechel R, Roessler W, Wieland WF, Filbeck T. Clinically relevant reduction in risk of recurrence of superficial bladder cancer using 5-aminolevulinic acid-induced fluorescence diagnosis: 8-year results of prospective randomized study. UROLOGY. 2007, 69, 675-679. (The tenth in the references list of revised manuscript)
  2. Roberts DW, Valdés PA, Harris BT, Fontaine KM, Hartov A, Fan X, Ji S, Lollis SS, Pogue BW, Leblond F, Tosteson TD, Wilson BC, Paulsen KD. Coregistered fluorescence-enhanced tumor resection of malignant glioma: relationships between delta-aminolevulinic acid-induced protoporphyrin IX fluorescence, magnetic resonance imaging enhancement, and neuropathological parameters. Clinical article. J NEUROSURG. 2011, 114, 595-603. (The eleventh in the references list of revised manuscript)
  3. Stummer W, Tonn JC, Mehdorn HM, Nestler U, Franz K, Goetz C, Bink A, Pichlmeier U; ALA-Glioma Study Group. Counterbalancing risks and gains from extended resections in malignant glioma surgery: a supplemental analysis from the randomized 5-aminolevulinic acid glioma resection study. Clinical article. J NEUROSURG. 2011, 114, 613-623. (The thirteenth in the references list of revised manuscript)
  4. Ishizuka M, Abe F, Sano Y, Takahashi K, Inoue K, Nakajima M, Kohda T, Komatsu N, Ogura S, Tanaka T. Novel development of 5-aminolevurinic acid (ALA) in cancer diagnoses and therapy. INT IMMUNOPHARMACOL. 2011, 11, 358-365. (The seventeenth in the references list of revised manuscript)
  5. Wiegell SR, Wulf HC. Photodynamic therapy of acne vulgaris using 5-aminolevulinic acid versus methyl aminolevulinate. J AM ACAD DERMATOL. 2006, 54, 647-651. (The eighteenth in the references list of revised manuscript)
  6. Zhang WF, Zhang F, Raziuddin R, Gong HJ, Yang ZM, Lu L, Ye QF, Zhou WJ. Effects of 5-Aminolevulinic Acid on Oilseed Rape Seedling Growth under Herbicide Toxicity Stress. J PLANT GROWTH REGUL. 2008, 27, 159-169. (The twenty-sixth in the references list of revised manuscript)
  7. Stummer W, Tonn JC, Goetz C, Ullrich W, Stepp H, Bink A, Pietsch T, Pichlmeier U. 5-Aminolevulinic acid-derived tumor fluorescence: the diagnostic accuracy of visible fluorescence qualities as corroborated by spectrometry and histology and postoperative imaging. NEUROSURGERY. 2014, 74, 310-319, 319-320. (The thirty-seventh in the references list of revised manuscript)

3: It will be informative to add the comparison on the Publication Counts in the Nanotechnology section between ALA and PpIX. The number of PpIX works is almost an order of magnitude larger.

Response: As the active form of 5-ALA, PpIX can be considered as the extension of 5-ALA applications. As suggested, we replotted the Figure 6A in the revised manuscript at page 21. The updated figure provides the publication information of 5-ALA-based and PpIX-based nanotechnology combination research, and their comparison simultaneously. Please find below the modified figure.

Figure 1 (Figure 6A in the revised manuscript). The trends of 5-ALA based and PpIX based nanomaterial combining annual publication numbers. Data were collected in the retrieve formula of TI = (*aminolevulinic acid OR 5 ALA) and TS = (nano*), and TS = (Protoporphyrin ix OR PpIX) and TS = (nano*) respectively.

4: In fact, I am disappointed by the uninformative conclusion that the main field of ALA research is photodiagnosis and photodynamic therapy. It is not the hotspots, there is nothing “hot”. It is just the common description for the most often ALA applications over many years; the terms allow distinguishing the ALA topic in the ocean of anticancer strategies and are routinely used by cancer researchers. Much more useful information can be obtained after “masking” the most common keywords and a closer analysis of the second level.

Response: It is indicated by the data analysis in the manuscript, photodiagnosis and photodynamic therapy was, and still is the predominate field of 5-ALA research. However, it is true that this field is common description for ALA applications over many years. To find more useful information for further developments in 5-ALA research, we performed a second level analysis after shielding the common keywords associated with photodiagnosis and photodynamic therapy, the produced landscape has showed in Supplementary Figure 3 in the revised manuscript. Those clusters including “derivative”, “in vitro”, “oxidative stress”, “photosynthesis”, “diabetes”, and “sonodynamic therapy” were newly emerged, which are also highly associated with the applications of 5-ALA. These new findings were also added into the revised manuscript at page 16 line 324-330. These are all highlighted in red in the main text, and it reads as below:

“Given the conception that photodiagnosis and photodynamic therapy is common description for ALA applications for years, we have performed a second level analysis after shielding the common keywords associated with photodiagnosis and photodynamic therapy (Supplementary Figure 1). The produced landscape showed that the clusters mainly associated with the applications of 5-ALA including “derivative”, “in vitro”, “oxidative stress”, “photosynthesis”, “diabetes”, and “sonodynamic therapy” were newly emerging.”

Figure 2 (Supplementary Figure 3 in the revised manuscript). The second level cluster analysis of keywords after shielding the common keywords associated with photodiagnosis and photodynamic therapy.

Minors

5: WoS is not the largest but one of the largest databases. Please, correct.

Response: Thank you for pointing out this inaccurate description, we have corrected it in revised manuscript. The change has been highlighted in red in the main text in Page 5 line 98-100, it now reads, “Web of Science (WoS, Clarivate Analytics, Philadelphia, PA, USA), one of the largest academic information databases, was taken as the sources to collect the related publications for following studies.”

6: Text in Fig. 1A is blind.

Response: Thanks for your reminder, we actually have mentioned the Figure 1A at page 5 line 123 in the “Materials and Methods” section.

7: The Publication number and calculated percent in Table 1 differ from the description in the main text.

Response: The differences have been corrected in the revised manuscript at page 7. However, there will be a slight difference of data retrieved from WoS though using the same formula at different timepoints. The article number hit in the day we submitted the manuscript was 1595 but the number in May 30th, 2022 was 1599. Considering the narrow range of the uncontrollable changes, conclusions will not be affected.

8: It is impossible to read any text in many Figures generated by CiteSpace. I offer to think on larger version, for example, in Supplementary Information.

Response: Thanks very much for your suggestion of supplying a large version of pictures. Please see the large versions of figures in the updated Supplementary Information.

9: The W. Stummer ranks in the list of the most cited articles are different in Table 3 and the main text.

Response: We have corrected this inconsistence in the revised manuscript at page 14. In fact, Stummer W holds the first, the second, and the fifth places in the list of top 10 most cited literatures, not the first, the third, and the fifth. The change has been highlighted in red in the main text in Page 14 line 277-278, it now reads, “Meanwhile, Stummer W holds the first, the second, and the fifth places in the list of top 10 most cited literatures (Table 3)”.

10: Table 4 summarizes not 15 but 30 keywords.

Response: We have checked and corrected it in the revised manuscript at page 17.

11: Please, thoroughly check using capital letters in the Figures and Tables, including those in Supplementary Information.

Response: We appreciate your careful reading. All the keywords are extracted from the original data, and they are shown as the original fonts as illustrated after the software running, this is also a common spelling way in other bibliometric papers [2-3]. But in tables, we have capitalized the first letter in the revised manuscript at page 17 in Table 4.

References:

  1. Developed Countries List 2022. World Population Review. 2022. Available online at: https://worldpopulationreview.com/country-rankings/developed-countries.
  2. Wu H, Cheng K, Guo Q, Yang W, Tong L, Wang Y, Sun Z. Mapping Knowledge Structure and Themes Trends of Osteoporosis in Rheumatoid Arthritis: A Bibliometric Analysis. Front Med (Lausanne). 2021, 23, 8:787228.
  3. Peng C, Kuang L, Zhao J, Ross AE, Wang Z, Ciolino JB. Bibliometric and visualized analysis of ocular drug delivery from 2001 to 2020. J Control Release. 2022, 345, 625-645.

Reviewer 2 Report

The paper from Y Zhou and co-workers is an interesting and timely investigation.

Comments

The authors never mention MAL (m-ALA) or HAL (h-ALA) despite these derivatives have considerably extended clinical 5-ALA applications in dermatology and urology. Is it done intentionally? MAL/HAL papers not always use 5-ALA as keyword. Please comment on that.  

The reference [22] of Peng et al (1997) should be replaced above, for instance together with references [5, 13, 14] on the same page.

The authors state on the page 3 that the growth rate of publications raised noticeably from 2019 compared with the past years. This observation is very interesting. What was the reason? Clinical applications or nanotechnology issues?

In the Conclusion section I would recommend to indicate one more time that the highlights of 5-ALA research and its derivative are related to fundamental biological effects as means of the improvement of efficacy.  

Author Response

Dear reviewer,

Thank you for kind consideration of our manuscript and offering the opportunity of major revision. We revised the manuscript according to your suggestions. The comments were addressed point by point, the changes made in the manuscript was highlighted. Please find the detailed responses below.

Comments

1: The authors never mention MAL (m-ALA) or HAL (h-ALA) despite these derivatives have considerably extended clinical 5-ALA applications in dermatology and urology. Is it done intentionally? MAL/HAL papers not always use 5-ALA as keyword. Please comment on that.

Response: Since methylaminolevulinate (MAL) and hexylaminolevulinate (HAL) could enhance the uptake of 5-ALA compared with the prototype, these two derivatives have gained their marketing authorization for a long time to overcome the limited local bioavailability of 5-ALA in clinical practice [1]. As a prodrug form of 5-ALA, it is reasonable to take them into consideration. In our present bibliometric analysis study, we collected the dataset with the retrieve formula of TI = (*aminolevulinic acid OR 5 ALA), which has included the TI = (methyl ester aminolevulinic acid) and TI = (hexyl ester aminolevulinic acid) indeed, we also validated it.  To make it clear, the sentences explaining this situation have now added in the introduction and conclusion sections. The relavent contents have been highlighted in red in the main text, they now read as below:

“Methyl ester aminolevulinic acid and hexyl ester aminolevulinic acid, two derivatives of 5-ALA, have also gained their marketing authorization to overcome the limited local bioavailability of 5-ALA in daily clinical practice [1]” at page 4 line 74-76.

“Given the conception that photodiagnosis and photodynamic therapy is common description for ALA applications for years [2], we have performed a second level analysis after shielding the common keywords associated with photodiagnosis and photodynamic therapy (Supplementary Figure 1). The produced landscape showed that the clusters mainly associated with the applications of 5-ALA including “derivative”, “in vitro”, “oxidative stress”, “photosynthesis”, “diabetes”, and “sonodynamic therapy” were newly emerging” at page 16 line 324-330.

“According to the cluster analysis and burst detection, the hotspots in this field are mainly concentrated on photodiagnosis and photodynamic therapy, while its derivatives and their fundamental biological effects were also emerging” at page 22 line 437-439.

2: The reference [22] of Peng et al (1997) should be replaced above, for instance together with references [5, 13, 14] on the same page.

Response: Thank you for this suggestion. We have adjusted the reference list. Please refer it in revised manuscript, the changes have been highlighted now in page 3 and page 4.

3: The authors state on the page 3 that the growth rate of publications raised noticeably from 2019 compared with the past years. This observation is very interesting. What was the reason? Clinical applications or nanotechnology issues?

Response: We refined out the photodiagnosis/photodynamic therapy related articles and nanomaterial related articles in the 5-ALA research dataset. As the figure showed, the publications of photodiagnosis/photodynamic therapy increased apparently since 2019. The increased number of publicayion in 2019, 2020, 2021 were 18, 38, 51, respectively, compared with the number of 2018, which is close to the growth of total publication (25, 41, 48 respectively). In contrast, nanotechnology related publications kept increasing steadily, the increased number of 2019, 2020, 2021 were 6, 0, 10 compared with 2018 respectively, which negligibly contributed to the growth of total publication (25, 41, 48 respectively). Therefore, accelerated growth rate since 2019 may mostly attributed to the photodiagnosis/photodynamic therapy related research. Additionally, the citation intensity in the co-citation cluster analysis (Figure 4B in the manuscript) also supported this conclusion. Because most of the connecting lines after 2019 were landed in photodiagnosis/photodynamic therapy related clusters.

The contents have been added in the main text as “Since most of the popular keywords and cluster were related with photodiagnosis and photodynamic therapy, we have drawn the annual changes photodiagnosis/photodynamic therapy related publications (Figure 4C). 911 publications were defined out and its growing path seems similar with the growth of total publications that raised apparently in 2019 compared with the past years, indicating that the major growth of publication in this area in last three years were from the photodiagnosis/photodynamic therapy related studies (Supplementary Figure 2)” at page 16 line 318-324.

4: In the Conclusion section I would recommend to indicate one more time that the highlights of 5-ALA research and its derivative are related to fundamental biological effects as means of the improvement of efficacy.

Response: Thanks for your good suggestion. We have added this point in the conclusion section.

The added content has been highlighted in red in the main text in Page 22 line 437-439, it now reads, “According to the cluster analysis and burst detection, the hotspots in this field are mainly concentrated on photodiagnosis and photodynamic therapy, while its derivatives and their fundamental biological effects were also emerging”.

References:

  1. Fotinos N, Campo MA, Popowycz F, Gurny R, Lange N. 5-Aminolevulinic acid derivatives in photomedicine: Characteristics, application and perspectives. Photochem Photobiol. 2006, 82(4), 994-1015.
  2. Ishizuka M, Abe F, Sano Y, Takahashi K, Inoue K, Nakajima M et al. Novel development of 5-aminolevurinic acid (ALA) in cancer diagnoses and therapy. INT IMMUNOPHARMACOL 2011, 11: 358-365.

Reviewer 3 Report

In the present manuscript, the authors perform a revision of the 5-ALA biomedical application, focusing in the analysis of scientists and areas of application. However, there are several issues that must be addressed before this manuscript can be considered for publication.

The manuscript fails to comply with the topics and subject areas described in the Pharmaceutics Aims and Scopes.

Some of the statistical analysis can be biased and with low significance.

Most of the times, more than the total amount of publications, the % that these publications represent on the total/overall of the country/institution would be more beneficial to find groups/institutions specialized in the application of 5-ALA.

Moreover, it would be interesting to compare the profile of increase in the 5-ALA publications with the development of nanomaterials field in general and more recently with the increase of interest in the photothermal and photodynamic therapies.

Author Response

Dear reviewer,

Thank you for kind consideration of our manuscript and offering the opportunity of major revision. We revised the manuscript according to your suggestions. The comments were addressed point by point, the changes made in the manuscript was highlighted. Please find the detailed responses below, and refer the enclosed file.

1: The manuscript fails to comply with the topics and subject areas described in the Pharmaceutics Aims and Scopes.

Response: This paper is a response to the invitation of special issue "Sustainable Materials and Technologies for Drug Delivery and Tissue Engineering". Theoretically, 5-ALA is natural non-protein amino acid as a common precursor of heme in animals, which indicated it is biologically safe. Meanwhile, the clinical applications and green synthesis developments of 5-ALA makes it sustainable. Moreover, 5-ALA was popular combing used in nano drug delivery systems nowadays. Therefore, a detailed bibliometric analysis to map the basic knowledge structure of this field, figuring out the hotspots and frontiers in the field of 5-ALA research in recent years will benefit to the future investigation of 5-ALA. Given all of that, we believe our manuscript is fully comply with the aims and scopes of Pharmaceutics, the editor invited and has already approved our abstract before this submission.

2: Some of the statistical analysis can be biased and with low significance.

Response: Web of Science (WoS) is one of the largest databases of academic information, therefore collecting data from WoS can ensure the data integrality. In the section of publication statistics, as mentioned that the growth rate of publication has raised significantly compared with the past years since 2019, by using linear regression model before and after 2018 (Figure 1), the raised growth rates of publications and citations were calculated. All the R values are larger than 0.85, which appears to be good fitting of the linear inclines. Some of the statistical analyses may be not significant in the subsequent section of “Knowledge structure features of 5-ALA research”, but they reflected the real state of 5-ALA research between 2006-2021. In addition, it should be noted that the modularity value (Q-value) and mean silhouette value (S-value) are two important parameters to evaluate the significance of community structure in cluster analysis, that is, Q > 0.3 and S> 0.7 is considered as a significant clustering [1]. All the cluster analyses have met these requirements in our present study, and the raw figures (Section: Large Version of CiteSpace Analysis) were added in the supplementary information for your reference.

Figure 1 (Supplementary Figure 1 in the revised manuscript). The linear regression model before and after 2018 in publication/citation terms. Data were processed using EXCEL 2016.

3: Most of the times, more than the total amount of publications, the % that these publications represent on the total/overall of the country/institution would be more beneficial to find groups/institutions specialized in the application of 5-ALA.

Response: In general, a percentage is more direct than an absolute value in demonstrating the comparisons. However, the percentage of a country/institution publications is calculated from the publication numbers, which means the results in our manuscript will not be changed using either forms of % or total amount to describe the publications. The percentages have also been provided (highlighted in red in the manuscript) in the table of major countries participating in 5-ALA research (Table 1 at page 7). As for the top active funding agencies, we also have listed the percentages in Supplementary Table 1.

4: Moreover, it would be interesting to compare the profile of increase in the 5-ALA publications with the development of nanomaterials field in general and more recently with the increase of interest in the photothermal and photodynamic therapies.

Response: Thanks for your excellent suggestion. As mentioned in our manuscript, 5-ALA is mainly used in photodiagnosis and photodynamic therapy, and nanotechnology combination is a potential direction in its application. The data search of 5-ALA studies in photothermal applications shows no relevant reports, indicating that it might not be a photothermal agent. Thus, we focused on the subfields of photodiagnosis/photodynamic therapy and nanotechnology combination. The figure below suggest that the photodiagnosis/photodynamic therapy related publications increased since 2019 and nanotechnology related publications were steadily in an increasing tendency, but they were approximately in the same level recently decade (Figure 2). Additionally, the result indicating that the accelerated growth rate of total publications since 2019 may mostly attributed to the photodiagnosis/photodynamic therapy related research.

Figure 2 (Supplementary Figure 2 in the revised manuscript). The publications comparison between nanomaterials related publications and photodiagnosis/photodynamic therapy related publications. Total publications were retrieved with the formula of TI = (*aminolevulinic acid OR 5 ALA); the nanomaterials related publications were collected in the retrieve formula of TS = (nano*) AND TI = (*aminolevulinic acid OR 5 ALA) or TS = (Protoporphyrin ix OR PpIX. the photodiagnosis/photodynamic therapy related publications were achieved in the retrieve formula of TI = (*aminolevulinic acid OR 5 ALA) AND TS = (photodiagnosis OR photodynamic). All publications were articles only.

References:

  1. Wu H, Cheng K, Guo Q, Yang W, Tong L, Wang Y et al. Mapping Knowledge Structure and Themes Trends of Osteoporosis in Rheumatoid Arthritis: A Bibliometric Analysis. Front Med (Lausanne) 2021, 8, 787228.

Round 2

Reviewer 1 Report

I thank the authors for taking into account all my comments. The manuscript can be published in its present form.